# Dysbiosis of Gut Microbiota and Intestinal Barrier Dysfunction in Pigs with Pulmonary Inflammation Induced by *Mycoplasma hyorhinis* Infection

Yingying Zhang,[a] Yuan Gan,[b] Jia Wang,[b] Zhixin Feng,[b] Zhaoxin Zhong,[c] Hongduo Bao,[a] Qiyan Xiong,[b] Ran Wang[a]

[a]Institute of Food Safety and Nutrition, Key Lab of Food Quality and Safety of Jiangsu Province-State Key Laboratory Breeding Base, Jiangsu Academy of Agricultural Sciences, Nanjing, China
[b]Key Laboratory for Veterinary Bio-Product Engineering, Ministry of Agriculture and Rural Affairs, Institute of Veterinary Medicine, Jiangsu Academy of Agricultural Sciences, Nanjing, China
[c]Jiangsu Coastal Area Institute of Agricultural Sciences, Yancheng, China

**ABSTRACT** Lung inflammation induced by *Mycoplasma hyorhinis* infection accounts for significant economic losses in the swine industry. Increasing evidence suggests that there is cross talk between the lungs and the gut, but little is known about the effect of the lung inflammation caused by *M. hyorhinis* infection on gut microbiota and intestinal barrier function. Here, we investigated changes in the fecal microbiotas of pigs with *M. hyorhinis* infection and the microbial regulatory role of such infection in intestinal barrier function. We infected pigs with *M. hyorhinis* and performed 16S rRNA gene sequencing analyses of fecal samples, data-independent acquisition (DIA) quantitative proteomic analyses of intestinal mucosa, and analyses of barrier dysfunction indicators in serum. We found that pigs with *M. hyorhinis* infection exhibit lung and systemic inflammation, as reflected by the histopathological changes and activation of the TLR4/MyD88/NF-$\kappa$B p65 signaling pathway in lung tissue, as well as the increased concentrations of serum inflammatory cytokines. Gut microbiotas tended to become disturbed, as evidenced by the enrichment of opportunistic pathogens. The increased diamine oxidase activities and D-lactate concentrations in serum and the decreased relative mRNA expression of *Occludin*, *ZO-1*, and *Mucin2* indicated the impairment of intestinal barrier function. Quantitative proteomic analyses showed a variety of altered proteins involved in immunomodulatory and inflammatory functions. There was a positive correlation between the abundance of opportunistic pathogens and inflammatory-cytokine concentrations, as well as intestinal immunomodulatory proteins. Our results suggest that lung inflammation induced by *M. hyorhinis* infection can contribute to the dysbiosis of gut microbiota and intestinal barrier dysfunction, and dysbiosis of gut microbiota was associated with systemic inflammation and intestinal immune status.

**IMPORTANCE** Cumulative evidence suggests that bacterial pneumonia may contribute to the dysbiosis of the gut microbiota and other gastrointestinal symptoms. Our experiment has demonstrated that lung inflammation induced by *M. hyorhinis* infection was associated with gut microbiota dysbiosis and intestinal barrier dysfunction, which may provide a theoretical basis for exploring the gut-lung axis based on *M. hyorhinis* infection.

**KEYWORDS** *Mycoplasma hyorhinis*, gut microbiota, proteomics, barrier function, gut-lung axis

Address correspondence to Hongduo Bao, baohongduo@163.com, Qiyan Xiong, qiyanxiongnj@163.com, or Ran Wang, dirkwang@126.com.

The authors declare no conflict of interest.

Respiratory tract diseases account for great economic losses in the swine industry worldwide and have become the most important health concern for swine producers today (1). *Mycoplasma hyorhinis*, a bacterium lacking cell walls in the class

*Mollicutes* that colonizes the upper respiratory tract (2), is a common cause of pneumonia, polyarthritis, serositis, eustachitis, and otitis (3). *M. hyorhinis* infection also leads to retarded piglet growth, reduced growth performance, and decreased feed utilization (4). Due to the prevalence of *M. hyorhinis* infection, growing bacterial drug resistance has complicated the treatment of pneumonia and systemic inflammation induced by *M. hyorhinis* infection (5). Therefore, there is a great need to learn more about the physiological and biochemical changes associated with the pneumonia caused by this pathogenic bacterium.

Accumulating evidence has demonstrated that gut microbes are associated with the onset and progression of disease (6, 7). It is now realized that the intestine plays a decisive role in promoting systemic inflammation and infection (8, 9). This is primarily because the large number and diverse population of microorganisms inhabiting in the intestinal tract interacts with the host and impacts mucosal and systemic immune function (10, 11). Pulmonary diseases are often accompanied by intestinal symptoms, the most prominent feature being shifts in gut microbiota composition and functions (12, 13). Recently, attempts have been made to modulate the gut microbiota by using probiotics and prebiotics to alleviate pulmonary diseases (14). These efforts indicated that there is a vital relationship between the two sites of the animal body. Emerging findings highlight the cross talk between the gut microbiota and the lungs, termed the "gut-lung axis." However, it is unclear whether *M. hyorhinis* infection can shape the gut microbiota and impact the host gut barrier function.

Based on the concept of the gut-lung axis, we hypothesized that lung inflammation induced by *M. hyorhinis* infection could change the gut microbiota composition and function and, furthermore, alter intestinal barrier function. Therefore, we inoculated pigs with *M. hyorhinis* and studied the resulting pulmonary injury, changes in gut microbiota, and changes in intestinal barrier function. The evidence obtained shows that the lung inflammation induced by *M. hyorhinis* infection can disrupt the gut microbiota, lead to functional changes in mucosal proteins, and then weaken the intestinal barrier function. Our results can further provide a theoretical basis for exploring the gut-lung axis based on bacterial infection.

## RESULTS

***M. hyorhinis* infection results in lung and systematic inflammation in pigs.** To first gain insight into the role of *M. hyorhinis* infection on lung inflammation, we infected pigs with *M. hyorhinis*. As expected, all the infected pigs exhibited histological evidence of severe lung injury. Compared with the control group, the infected pigs showed more inflammatory effects in their hematoxylin-and-eosin (HE)-stained lung sections, including significant interstitial infiltration of inflammatory cells and thickening of the alveolar walls (Fig. 1A). As the TLR4 (Toll-like receptor 4)/MyD88/NF-$\kappa$B p65 signaling pathway is directly involved in the immune responses and inflammation (15, 16), we used immunohistochemistry to determine levels of TLR4, MyD88, and NF-$\kappa$B p65 protein expression in paraffin-embedded lung sections. The results showed that the percentages of cells staining positive for TLR4, MyD88, and NF-$\kappa$B p65 in the *M. hyorhinis* infection group (MH) were higher than those in the control group (Fig. 1B to D). Meanwhile, the concentrations of TLR4, MyD88, and NF-$\kappa$B in serum of *M. hyorhinis*-infected pigs were also increased (Table 1) ($P < 0.05$). These results demonstrate that *M. hyorhinis* can activate the TLR4/MyD88/NF-$\kappa$B pathway. When pathogens invade the body, it generally mounts an inflammatory response (17). Therefore, we measured the serum cytokine levels. It can be seen from the data in Table 2 that the treated pigs exhibited higher tumor necrosis factor alpha (TNF-$\alpha$) ($P < 0.05$) and interleukin 6 (IL-6) ($P = 0.058$) concentrations than the controls. To assess the severity of growth retardation in *M. hyorhinis*-infected pigs, the initial and final body weights of all pigs were recorded. *M. hyorhinis* treatment led to a significant drop in average daily weight gain (Table 3) ($P < 0.05$).

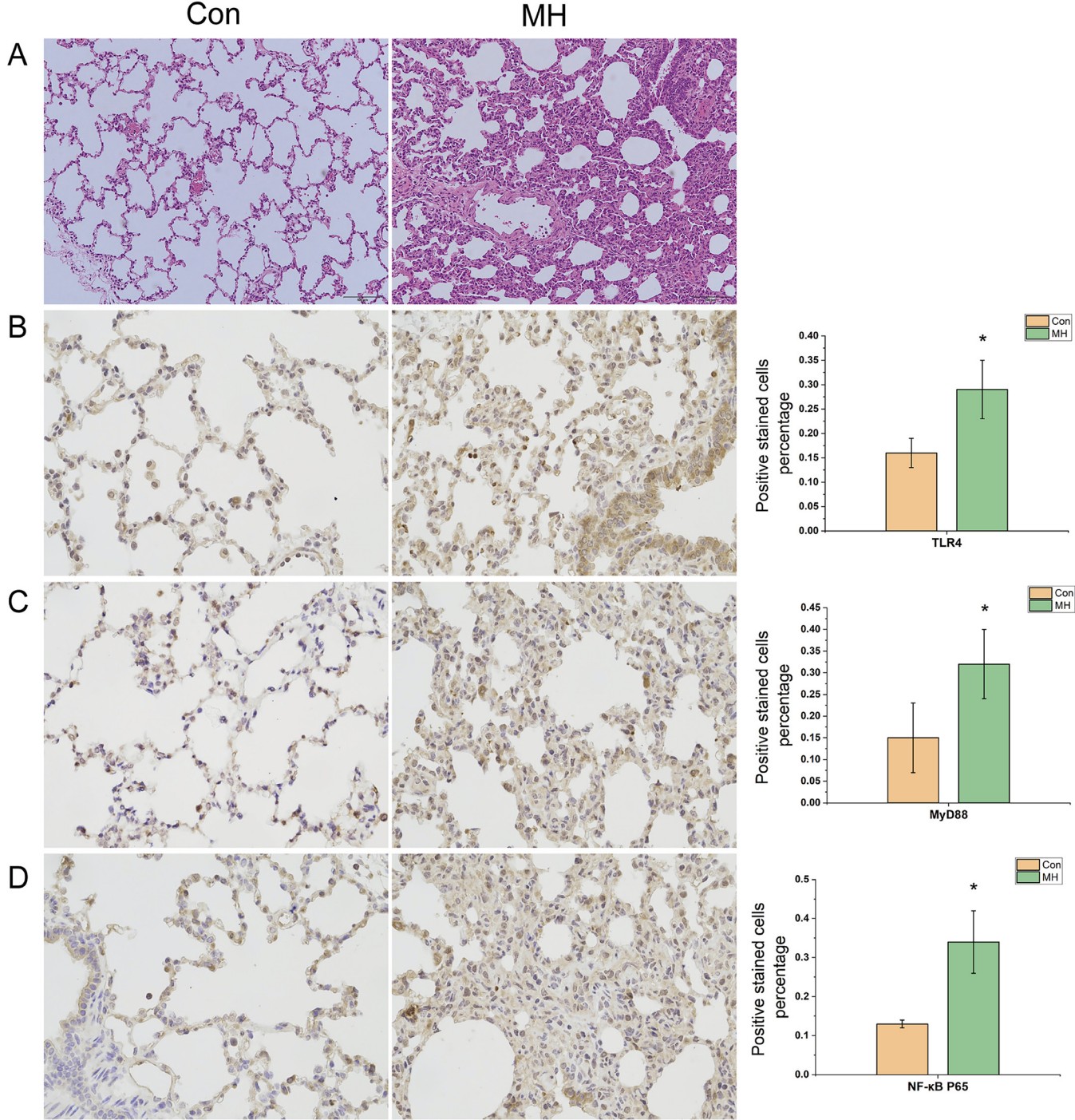

**FIG 1** (A) Representative slides of lung sections from pigs infected with *M. hyorhinis*, taken to assess pulmonary inflammation. H&E staining; magnification, ×100. (B to D) Effect of *M. hyorhinis* infection on percentages of TLR4-, MyD88-, and NF-κB-positive stained cells.

**Lung inflammation induced by *M. hyorhinis* infection can contribute to the dysbiosis of gut microbiota in pig.** To determine whether lung inflammation induced by *M. hyorhinis* infection is associated with the gut microbiota, we analyzed the gut microbiota composition in pigs. All pigs' microbiota characteristics are displayed in Fig. 2. The rarefaction curve tends to be flat, indicating that the sequencing depth is sufficient. However, the sample curve of the *M. hyorhinis*-infected (MH) group is higher than that of control group, indicating that there are many more species in *M. hyorhinis*-treated pigs (Fig. 2A). Gut microbiota diversity was assessed using different diversity

**TABLE 1** TLR4, MyD88, and NF-$\kappa$B concentrations in serum of *M. hyorhinis*-infected pigs

| Protein | Concn (ng/mL) in group[a] | | P value |
| --- | --- | --- | --- |
| | Control | MH | |
| TLR4 | 1.03 ± 0.11 | 1.85 ± 0.32 | 0.003 |
| MyD88 | 3.17 ± 0.47 | 5.24 ± 1.11 | 0.011 |
| NF-$\kappa$B | 4.66 ± 0.98 | 10.12 ± 1.41 | <0.001 |

[a]Results are means ± standard deviations (SD) ($n = 6$).

indexes, including observed, Chao1, ACE, Shannon, Simpson, and coverage indexes. As shown in Fig. 2B, gut microbial diversity, as estimated by observed, Chao1, and ACE, was greater in MH than in controls ($P < 0.05$). This showed that the *M. hyorhinis*-treated pigs attained higher levels of bacterial richness. The principal-coordinate analysis (PCoA) plots based on UniFrac distances revealed a separation of the two group samples (Fig. 2C), demonstrating that the microbial composition of the two groups differed significantly. In terms of alterations in the microbial communities at the phylum level (Fig. 2D), a total of 17 dominant phyla were identified in each group. *Firmicutes* (76.29% versus 54.78%), *Proteobacteria* (2.49% versus 0.92%), and *Tenericutes* (0.14% versus 0.03%) were enriched in *M. hyorhinis*-treated pigs ($P < 0.05$), whereas *Spirochaetes* (0.09% versus 4.67%) and *Bacteroidetes* (19.23% versus 37.19%) were depleted in *M. hyorhinis*-treated pigs ($P < 0.05$). The differential genera are shown in Fig. 2E; 18 genera were found to be significantly different between the two groups. Of these discriminatory taxa, *Romboutsia* ($P = 0.00031$), *Desulfovibrio* ($P = 0.00035$), *Alistipes* ($P = 0.00102$), *Anaerovorax* ($P = 0.00142$), *Lachnospiraceae incertae sedis* ($P = 0.00226$), *Anaerobacterium* ($P = 0.0057$), and *Lactobacillus* ($P = 0.00808$) were found to be more abundant in the *M. hyorhinis*-treated pigs.

**Intestinal barrier dysfunction during lung inflammation induced by *M. hyorhinis* infection.** Imbalances in the gut microbiota may lead to impairment of the intestinal morphology and barrier function (18, 19). Therefore, we visualized the villus height (VH) and crypt depth (CD) of the intestine in all pigs. The pigs in the MH group exhibited lower intestinal VH and VH/CD, but higher intestinal CD, than pigs in the control group (Fig. 3A; Table 4) ($P < 0.01$). Diamine oxidase (DAO) and D-lactate are commonly used as markers of intestinal barrier dysfunction (20). To examine whether lung injury induced by *M. hyorhinis* infection can impair intestinal barrier function, we measured serum DAO activity and D-lactate concentration. The results showed that DAO activities and D-lactate concentrations in MH group were higher than those in the control group (Fig. 3B and C). As claudins, occludins, and zona occludens proteins (ZOs) are central to the regulation of tight-junction permeability (20), *Claudin-1*, *Occludin*, *ZO-1*, and *ZO-2* genes in the intestinal mucosa were examined by qPCR. The results show that the weakened barrier function correlated with decreased relative mRNA expression of *Occludin*, *ZO-1*, and *Mucin2* (Fig. 3D to H).

**Lung inflammation induced by *M. hyorhinis* infection is accompanied by variations in gut proteomics.** To further evaluate the molecular alterations in the gut, the intestinal mucosal proteome was assessed by quantitative proteomics. A total of 7,162 proteins were identified in the intestinal mucosa. A Venn diagram showed that 77 proteins were unique to the untreated pigs and 255 proteins were unique to the *M. hyorhinis*-treated pigs (Fig. 4A). The principal component analysis (PCA) revealed a clear separation

**TABLE 2** Effect of *M. hyorhinis* infection on serum cytokine levels

| Cytokine | Concn (ng/L) in group[a] | | P value |
| --- | --- | --- | --- |
| | Control | MH | |
| TNF-$\alpha$ | 7.21 ± 1.57 | 19.71 ± 5.67 | 0.021 |
| IL-1$\beta$ | 0.70 ± 0.14 | 0.99 ± 0.41 | 0.237 |
| IL-6 | 211.91 ± 39.21 | 278.71 ± 25.23 | 0.058 |

[a]Results are means ± standard deviations (SD) ($n = 6$).

**TABLE 3** Body weight of *M. hyorhinis* infected pigs

| Wt measurement | Value in group[a] | | *P* value |
| --- | --- | --- | --- |
| | Control | MH | |
| Initial wt (kg) | 6.06 ± 1.21 | 5.51 ± 1.39 | 0.549 |
| Final wt (kg) | 7.68 ± 0.91 | 6.20 ± 1.62 | 0.126 |
| Avg daily wt gain (kg/day) | 0.07 ± 0.03 | 0.03 ± 0.01 | 0.036 |

[a]Results are means ± standard deviations (SD) (*n* = 6).

between the control and MH samples, indicating that the protein species of the two groups differed significantly (Fig. 4B). Quantitative analysis revealed that 391 proteins were significantly changed. Among them, 141 proteins were upregulated, whereas 250 proteins were downregulated (Fig. 4C). Box plots of some significantly altered proteins are shown in Fig. 4D.

Functional annotation of the upregulated cluster revealed proteins involved in complement and coagulation cascades, lysosome, oxidative phosphorylation, insulin secretion, DNA replication, inositol phosphate metabolism, the phosphatidylinositol signaling system, protein digestion and absorption, cardiac muscle contraction, cell cycle, axon guidance, and ribosome biogenesis in eukaryotes. Functional annotation of the downregulated cluster revealed these proteins to be involved in fat digestion and absorption, alcoholic liver disease, the adipocytokine signaling pathway, protein processing in the endoplasmic reticulum, glycerophospholipid metabolism, the NOD-like receptor signaling pathway, drug metabolism-cytochrome P450, renin secretion, pyrimidine metabolism, arginine and proline metabolism, metabolism of xenobiotics by cytochrome P450, and circadian rhythm (Fig. 4E).

**Systemic inflammation and gut proteomic alterations are associated with gut microbiota dysbiosis.** In order to detect interactions between the serum cytokines, altered genera, and differential proteins, a Pearson correlation analysis was carried out. Figure 5 shows a range of correlation coefficients for the serum cytokines and the altered genera and proteins, ranging from 1.0 (maximum positive correlation) to −1.0 (maximum negative correlation), with 0 indicating no correlation. Genera upregulated by *M. hyorhinis* infection, including *Lactobacillus*, *Lachnospiraceae incertae sedis*, *Romboutsia*, *Anaerovorax*, *Escherichia/Shigella*, *Desulfovibrio*, *Blautia*, *Alistipes*, and *Catabacter*, were positively correlated with TNF-$\alpha$ and IL-6 concentrations, while genera downregulated by *M. hyorhinis* infection, including *Parabacteroides*, *Treponema*, *Intestinibacter*, and *Anaerovibrio*, were negatively correlated with TNF-$\alpha$ and IL-6 concentrations (Fig. 5A). Meanwhile, upregulated genera were positively correlated with some differential proteins (GSTA2, GPAT3, TMEM167A, dipeptidyl peptidase 8 isoform 3, RBM45, ADA, RBP2, interferon-induced GTP-binding protein Mx2, and interferon-induced GTP-binding protein Mx1) and negatively correlated with others (MATK, SMOTH, CNN1, CALHM6, JCHAIN, ROMO1, SUOX, Ig-like domain-containing protein, and LIMA1) (Fig. 5B).

## DISCUSSION

Respiratory disease is a major health concern in the pig industry worldwide (21). Among the most frequently detected pathogens, *Mycoplasma hyopneumoniae* is the main pathogen causing pneumonia and has been studied intensively (22). However, the pathological changes in pigs caused by *M. hyorhinis* infection are less well understood. Although *M. hyorhinis* has been proposed as a possible primary pathogen causing pneumonia (23, 24), it can also cause other inflammatory porcine diseases (25–27) and coinfection with a variety of pathogens (28–30). In the present study, we confirmed the effect of *M. hyorhinis* infection individually on the lung inflammation profile of pigs. Meanwhile, we showed that the lung inflammation induced by *M. hyorhinis* infection can contribute to dysbiosis of the gut microbiota and intestinal barrier dysfunction, as evidenced by the increased abundance of pathogenic bacteria and shifts

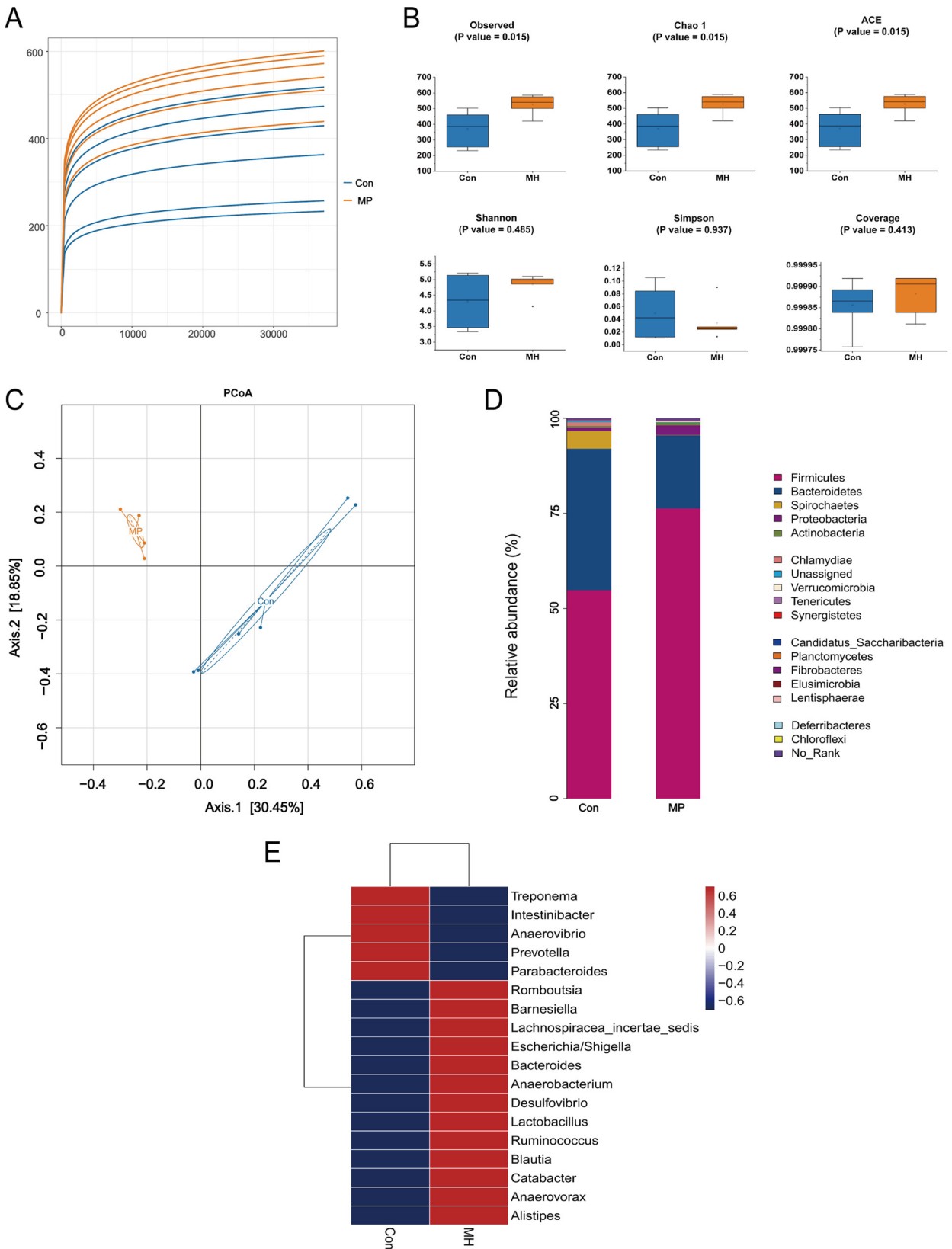

**FIG 2** (A) Rarefaction curves of operational taxonomic units (OTUs) in fecal samples. (B) The alpha diversity was estimated based on the observed, Chao1, ACE, Shannon, Simpson, and coverage indexes. (C) PCoA of bacterial communities based on UniFrac distances. (D) Relative abundance of fecal microbial communities at the phylum level. (E) Heat map of significantly altered genera (red represents a significant increase in abundance, and blue represents a significant decrease in abundance).

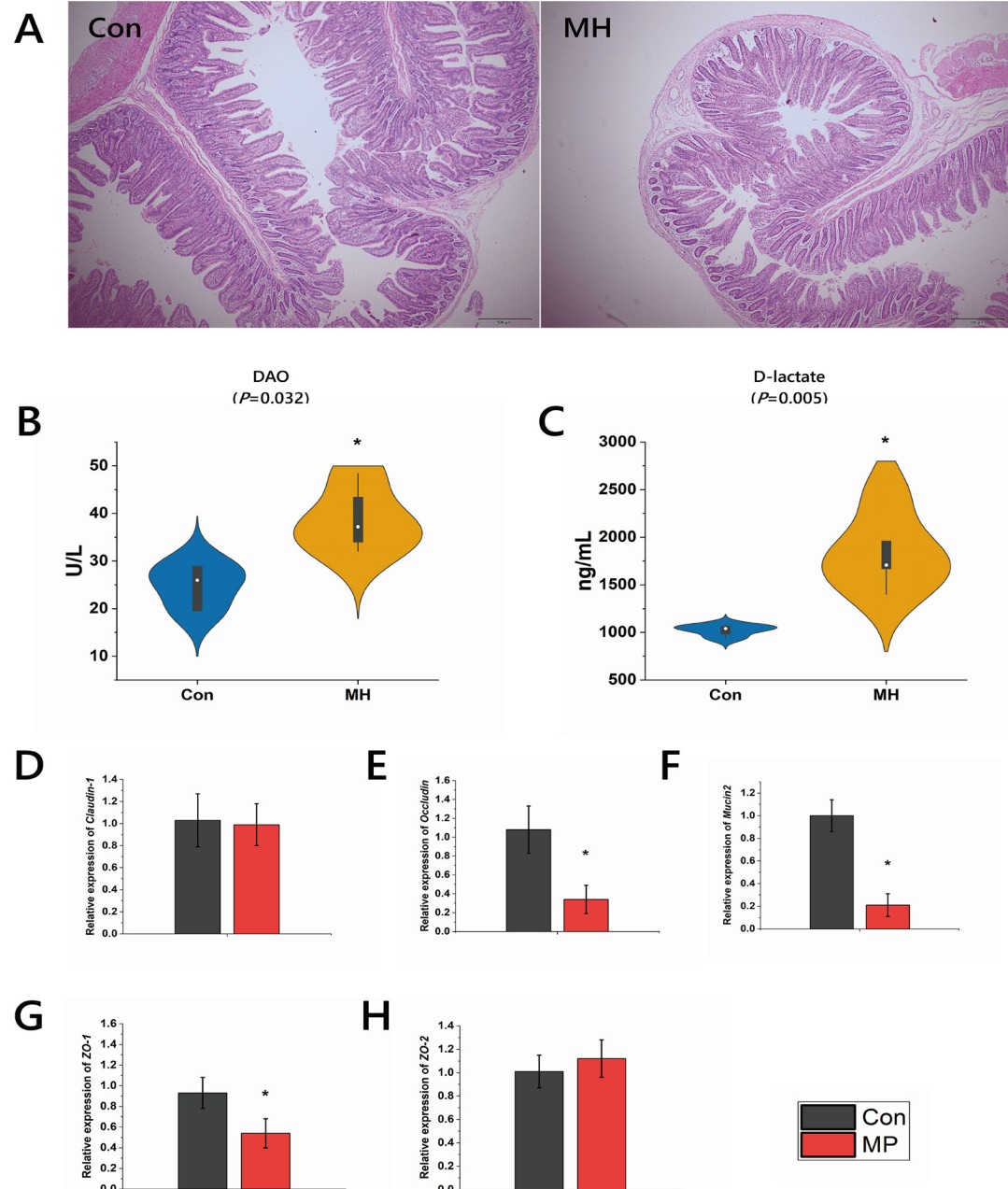

**FIG 3** (A) Morphological changes of jejunum tissue of pigs infected with *M. hyorhinis*. H&E staining; magnification, ×100. (B and C) Diamine oxidase activities and D-lactate concentrations in serum. (D to H) Relative mRNA expressions of *Claudin-1*, *Occludin*, *Mucin2*, *ZO-1*, and *ZO-2* in intestinal mucosa.

in intestinal mucosal proteomics, as well as the increased concentrations of gut barrier dysfunction markers.

This study emphasizes the concepts that the lung and the gut are intricately connected and that pathogen-induced lung inflammation may alter the gut microbiota composition. In particular, some recent breakthrough discoveries reinforced this view. Zuo et al. demonstrated that patients with COVID-19 had significant alterations in gut microbiota, mainly increased abundance of opportunistic pathogens and depletion of beneficial commensals (31). Similarly, people with chronic obstructive pulmonary disease were found to have varieties of bacterial species differing from healthy individuals in their fecal microbiome. Some disease-associated bacteria are associated with weakened lung function (32). Consistent with these reports, the present study also found

**TABLE 4** Intestinal morphology of *M. hyorhinis* infected pigs

| Items | Value in group[a] | | P value |
|---|---|---|---|
| | Control | MH | |
| VH ($\mu$m) | 482.71 ± 74.90 | 362.56 ± 59.22 | <0.001 |
| CD ($\mu$m) | 150.70 ± 38.94 | 192.07 ± 35.07 | 0.002 |
| VH/CD | 3.34 ± 0.98 | 1.95 ± 0.51 | <0.001 |

[a]Results are means ± standard deviations (SD) ($n = 6$).

that pigs infected with *M. hyorhinis* had enrichment of some pathogens and opportunistic pathogens, including *Alistipes*, *Blautia*, *Ruminococcus*, *Desulfovibrio*, *Escherichia/Shigella*, and *Bacteroides*. Most of these bacteria are intestinal proinflammatory bacteria (33–38), indicating that lung injury may trigger intestinal inflammation.

We also identified an upregulated *Firmicutes/Bacteroidetes* ratio at the phylum level. Previous studies tended to suggest that an upregulated *Firmicutes/Bacteroidetes* ratio acts as an indicator of some pathological conditions and gut dysbiosis (39, 40). These results lead us to believe that *M. hyorhinis* infection can contribute to dysbiosis of the gut microbiota. Most current research suggests that lung infection can contribute to the dysbiosis of the gut microbiota, because lung infection mainly alters the gut microbiota by affecting the inflammatory immune response. For example, patients with coronavirus disease 2019 (COVID-19) have been found to have some adverse immune responses, including increased proinflammatory cytokines and lymphocytopenia (41, 42). Excessive proinflammatory cytokine circulation can alter the gut microbiota and disturb intestinal integrity. In our present study, we found that pigs with *M. hyorhinis* infection had evidence of lung inflammation, including significant interstitial infiltration of inflammatory cells and thickening of the alveolar walls, as well as the activation of the TLR4/MyD88/NF-$\kappa$B p65 signaling pathway. It is recognized that activation of TLR4/MyD88/NF-$\kappa$B p65 can cause the expression of proinflammatory cytokines to increase. We did find higher levels of proinflammatory cytokines in the serum of the infected pigs. Therefore, we infer that cytokine modulator may be the link between lung and gut, but much work remains to be done to confirm this.

A potential finding of our study is that the disturbed microbiota may drive intestinal gut barrier dysfunction and inflammation, which was evidenced by the increased D-lactate concentrations and DAO activities in serum, as well as the downregulated relative mRNA expressions of *Occludin*, *ZO-1*, and *Mucin2* and the shifted proteomics. D-Lactate is the metabolite of bacterial fermentation. Like DAO, the concentration of serum D-lactate increase when the gut epithelial barrier experiences dysfunction (20). Evidence has been accumulating in recent years emphasizing the involvement of gut microbiota dysbiosis in gut barrier dysfunction. The dysregulated microbiota is generally characterized by a reduced abundance of butyric acid-producing bacteria, which have obvious proinflammatory effects (19). As mentioned above, some proinflammatory bacteria were also found in our results, such as *Alistipes*, *Blautia*, *Ruminococcus*, *Desulfovibrio*, *Escherichia/Shigella*, and *Bacteroides*. They were all positively correlated with the elevated TNF-$\alpha$ and IL-6 concentrations, suggesting that the increased abundance of these genera does contribute to systemic inflammation.

Gut barrier dysfunction indicates some possible changes in functional intestinal protein molecules. Therefore, the proteome profiling of the mucosa is used to explore molecular alterations in the gut. Of the significantly increased proteins, several have known immunomodulatory functions, including JCHAIN (joining chain of multimeric IgA and IgM), IGHG (IgG heavy chain), and Ig-like domain-containing protein. The Ig-like domain-containing protein is a typical feature of proteins belonging to the immunoglobulin superfamily (43). The mammalian gut has five classes of immunoglobulins, IgA, IgG, IgM, IgE, and IgD. Of these, IgA can bind to antigens from food and microbiota, thus preventing their invasion into host intestinal epithelial cells (44, 45). Overproduction of these immunoglobulins supports our hypothesis that dysbiosis of gut microbiota may

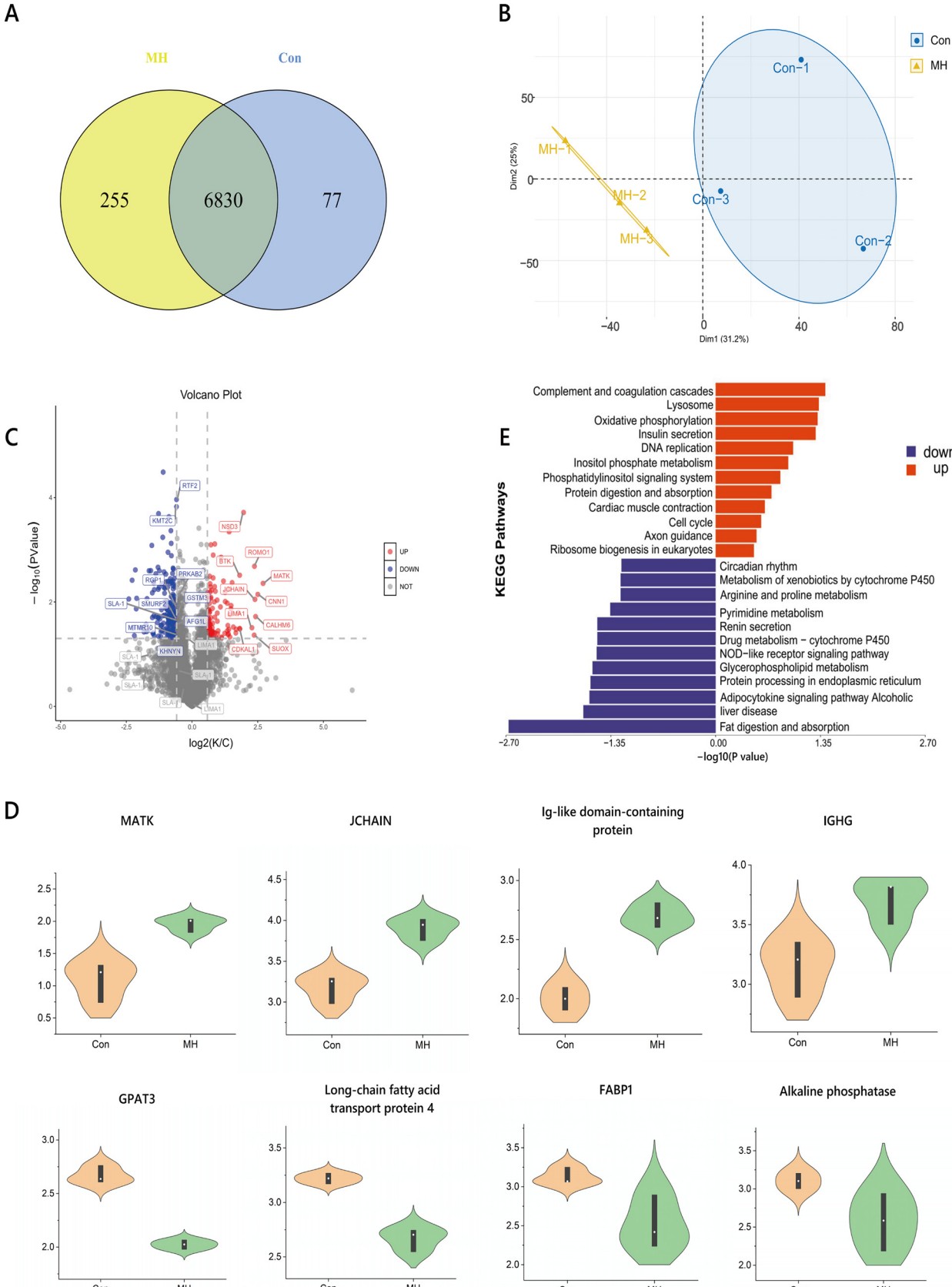

**FIG 4** (A) Venn diagram analysis of overlapping and differential proteins. (B) PCA. (C) Volcano plot showing significantly up- and downregulated proteins. (D) Box plots of some differential proteins. (E) KEGG pathway enrichment analysis of differentially expressed proteins.

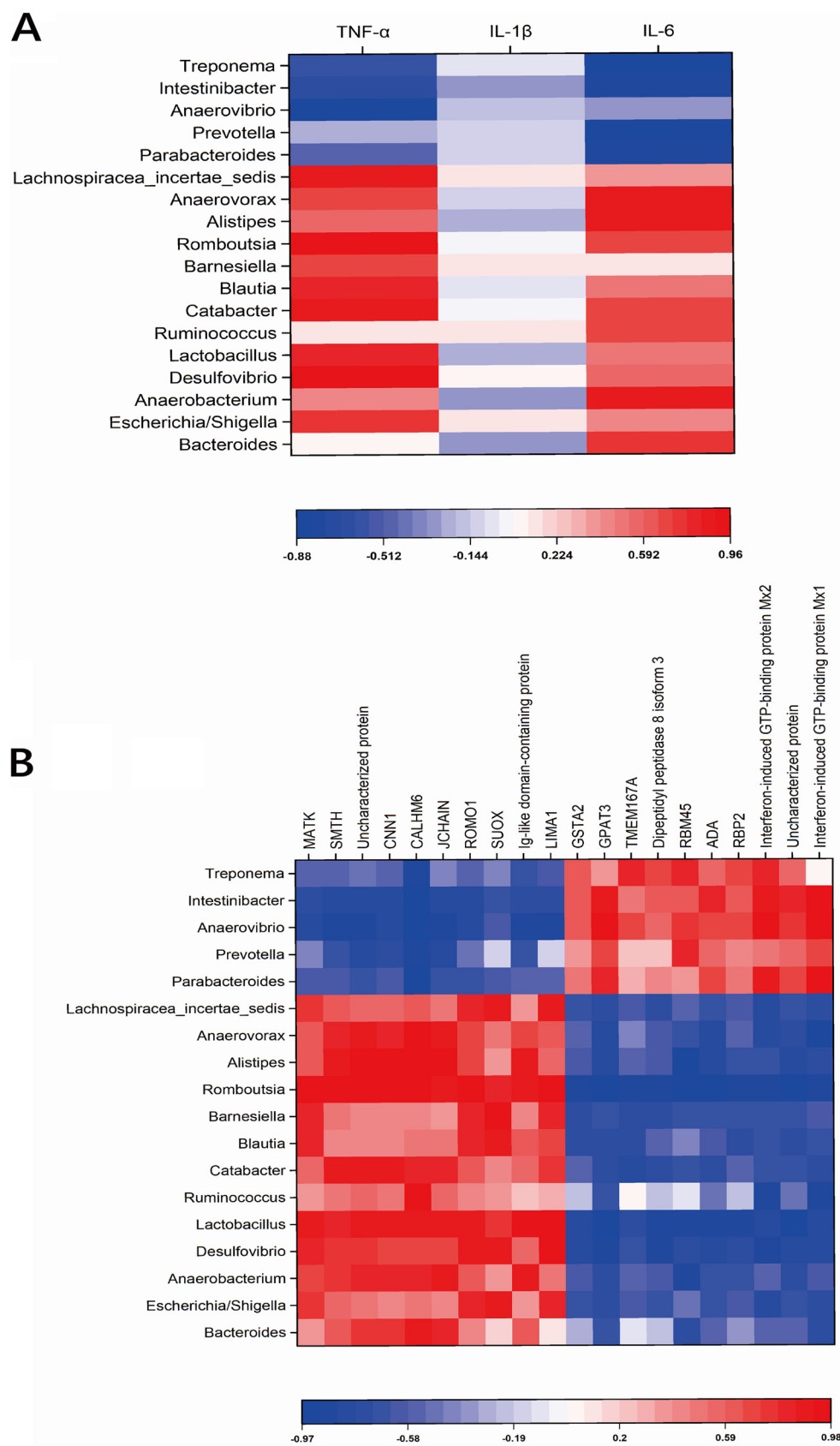

**FIG 5** (A) Pearson's correlations between altered fecal genera and serum cytokines. (B) Pearson's correlations between altered fecal genera and differentially expressed proteins. Colors represent correlations, with positive correlations in red and negative correlations in blue.

trigger intestinal inflammation and lead to immune dysregulation. In addition, another increased protein, MATK (tyrosine-protein kinase), may be involved in inflammatory and immune regulatory pathways. It is a key signaling molecule and important regulator of processes closely linked to the NF-$\kappa$B signaling pathway (map04064), epithelial cell signaling in *Helicobacter pylori* infection (map05120), and the B cell receptor signaling pathway (map04662) (https://www.kegg.jp/entry/K05854). The increased MATK may activate the gut immune status via those signaling pathways. Of the significantly decreased proteins, alkaline phosphatase is expressed in the enterocytes of the proximal small intestine, exerts an anti-inflammatory effect by detoxifying a variety of bacterial proinflammatory factors, such as lipopolysaccharide (LPS), CpG-DNA, and flagellin (46, 47), and can enhance barrier function by upregulating tight-junction proteins (48). Unfortunately, the decrease in abundance of alkaline phosphatase seen in this study would aggravate the weakening of barrier function. In addition, we observed that some proteins decreased are mainly involved in the process of lipid metabolism, such as FABP1 (fatty acid-binding protein), long-chain fatty acid transport protein 4, and GPAT3 (glycerol-3-phosphate acyltransferase 3). It is well established that the gut microbiota can affect lipid metabolism through metabolites such as bile acids and short-chain fatty acids, which enables it to regulate lipid absorption and clearance (49). These observations suggest that gut microbiota dysbiosis may further lead to changes in lipid metabolism.

In summary, our work highlights several distinctive features of pulmonary inflammation induced by *M. hyorhinis* infection in pigs, including increased gut microbial diversity, enriched abundance of pathogens and opportunistic pathogens, altered mucosal proteins with immunomodulatory function, and weakened intestinal barrier function (Fig. 6). These results will provide an important molecular basis for understanding the gut symptoms of lung *Mycoplasma* infection and will shed light on the cross talk between the lungs and gut microbiota.

## MATERIALS AND METHODS

**Ethical approval.** All animal work was carried out according to the approved guidelines established by the Ministry of Agriculture of China. All experimental design and procedures were approved by the Institutional Animal Care and Use Committee of Jiangsu Academy of Agricultural Sciences (ID: XYSK 2015-0020).

**Animal and experimental design.** A total of 12 male Bama miniature pigs 2 to 2.5 months old were purchased from a commercial piggery (Zhoubang Biological Technology Company of Nanjing, Nanjing, China). Pigs were determined to be free of colonization by *M. hyorhinis* and free of antibodies to *M. hyorhinis*. All healthy pigs were weighed and randomly assigned to two treatment groups: (i) the control group and (ii) the MH group (the *M. hyorhinis* infection group). All pigs were housed in individual pens and allowed free access to water and feed for the 42-day experiment. The basal diets were formulated to meet the nutritional requirements stated by the National Research Council (NRC; 2012) and are presented in Table S1 in the supplemental material. The isolation and preparation of *M. hyorhinis* were as described in previous literature (50). Challenge doses were based on the study by Martinson et al. with slight modifications and quantified by color-changing units (CCU) (3). Pigs were inoculated with 20 mL intraperitoneally on day 19, 10 mL intravenously on day 20, and 10 mL intranasally on day 21 for a total dose of $1 \times 10^9$ CCU per animal. Pigs in the control group were given an equivalent volume of saline by the same inoculation method as the pigs in the MH group.

**Sample collection.** On day 42, all pigs were weighed, then stunned electrically, and slaughtered immediately via exsanguination of the left carotid artery. Approximately 10-mL blood samples were collected from each pig and immediately centrifuged at $3,000 \times g$ for 10 min to obtain serum. After evisceration, the lungs and digestive tracts were removed, and approximately 1 cm$^2$ of the left lung middle lobe and 1-cm jejunum segments were fixed in 4% phosphate-buffered paraformaldehyde for morphological analysis. The mucosae of jejunal were gently scraped. The feces of each pig were collected separately, mixed evenly, and then sampled. All mucosa and feces samples were stored at $-80°$C for further analysis.

**Histomorphological analysis.** Each sample of lung was processed and embedded in paraffin to make formalin-fixed paraffin-embedded blocks. Hematoxylin and eosin (HE) staining was performed for histomorphological analysis. Images were obtained with a digital camera (Olympus Optical Co. Ltd., Tokyo, Japan). The VH (in micrometers, from the tip of villus to the villus-crypt junction level for 10 villi per section) and CD (in micrometers, the vertical distance from the villus-crypt junction to the lower limit of the crypt for 10 corresponding crypts per section) were analyzed by Image-Pro Plus 6.0 software (Media Cybernetics, Bethesda, MD, USA).

**Immunohistochemical analysis of paraffin sections.** Paraformaldehyde-fixed, paraffin-embedded specimens of lung tissues were examined to evaluate expression levels of TLR4, MyD88, and NF-$\kappa$B proteins, as follows. The sections were deparaffinized in xylene and rehydrated through a graded alcohol series. Next, the sections were pretreated in a microwave oven, and peroxidase activity was blocked with 3% hydrogen peroxide for 20 min, followed by incubation with primary antibody to TLR4 (GB11519;

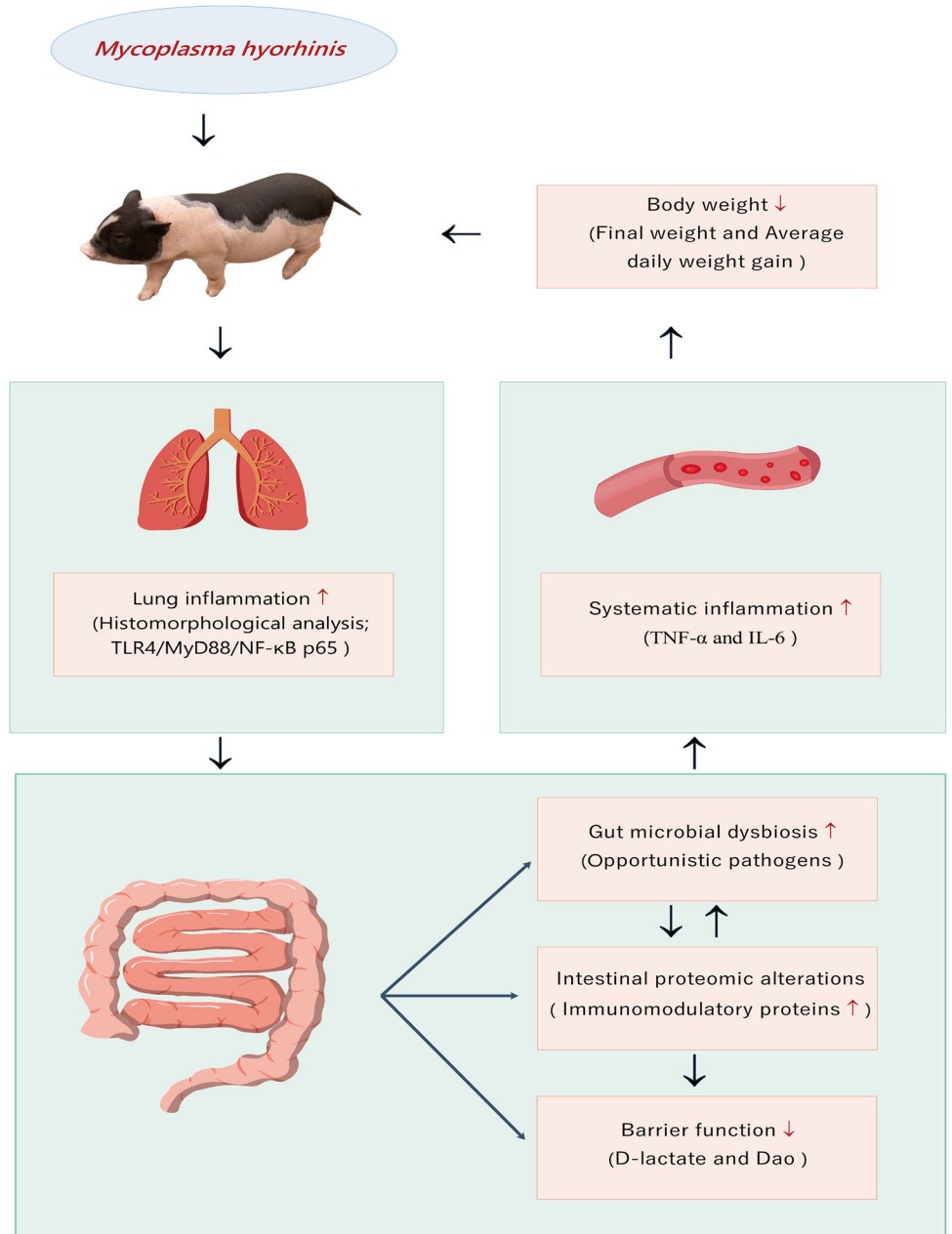

**FIG 6** Systematic analysis of the effects of pulmonary inflammation induced by *Mycoplasma hyorhinis* on gut microbiota, intestinal barrier function, intestinal mucosal proteomics, serological cytokines, and growth performance.

Servicebio Co. Ltd., Wuhan, China), MyD88 (4283S; Cell Signaling Technology, Inc., Beverly, MA, USA), and NF-$\kappa$B (CY2329; Abways Technology Inc, Shanghai, China) overnight at 4°C. Then, sections were washed and incubated with secondary antibody (horseradish [HRP]-conjugated goat anti-mouse IgG for immunohistochemistry [IHC]; Bio Basic Canada, Inc., Markham, ON, Canada). Staining was performed using a diaminobenzidine (DAB) kit (DA1015, Solarbio Co. Ltd., Wuhan, China). The percentage of positively stained cells was determined by obtaining images with a digital camera (Olympus Optical Co. Ltd., Tokyo, Japan) and analyzing them with Image-Pro Plus 6.0 software (Media Cybernetics, Bethesda, MD, USA). Approximately 50 cells in each image were evaluated. The percentage of positive cells for each image was calculated as the number of positive cells divided by the total number of cells counted, multiplied by 100. Four different images were selected for each lung section, and the average value was used for statistical analysis.

**Serum assays.** The kits of IL-1$\beta$ (catalogue no. H002), IL-6 (catalogue no. H007-1-2), TNF-$\alpha$ (catalogue no. H052-1), TLR4 (catalogue no. H449-1), NF-$\kappa$B (catalogue no. H202), D-lactate (cata-logue no. H263-1-1) and diamine oxidase (catalogue no. A088-1-1) were purchased from Nanjing Jiancheng Bioengineering Institute, so these catalogue numbers in this sentence are Nanjing Jiancheng Bioengineering Institute numbers. The MyD88 (catalogue no. ML05553) was purchased from Mlbio

Biotechnology Co. Ltd, so the catalogue no. ML05553 was Mlbio Biotechnology number. The inter- and intra-assay coefficients of variation (CV) obtained with diagnostic enzyme-linked immunosorbent assay (ELISA) kits for TNF-$\alpha$, IL-1$\beta$, IL-6, TLR4, NF-$\kappa$B, and D-lactate are <10% and <12%. The inter- and intra-assay CV for MyD88 are <10% and <15%. The detection limits were as follows: TNF-$\alpha$, 3 to 900 ng/L; IL-1$\beta$, 0.5 to 200 ng/L; IL-6, 2 to 600 ng/L; TLR4, 0.05 to 20 ng/mL; MyD88, 0.25 to 8 ng/mL; NF-$\kappa$B, 0.05 to 20 ng/mL; and D-lactate, 20 to 6,000 ng/mL.

**RNA extraction and qPCR analysis.** Total RNA was extracted using the TRIzol reagent (Thermo Fisher Scientific, Waltham, MA, USA); then, the isolated RNA was quantified with a Nanodrop ND-1000 spectrophotometer (Thermo Fisher Scientific, Wilmington, DE, USA). Total RNA was reverse transcribed to cDNA using reverse transcription (RT) master premix (TransGen Biotech Co., Ltd., Beijing, China). The RT reaction mixtures were incubated for 15 min at 42°C, followed by 5 s at 85°C. The RT products (cDNA) were stored at −20°C. qPCR was performed using PerfectStart Green qPCR SuperMix (TransGen Biotech Co., Ltd., Beijing, China) on the LightCycler 480 system (Roche, Basel, Switzerland). Briefly, an amplification reaction was carried out in a total volume of 20 $\mu$L, containing 10 $\mu$L of 2× PerfectStart Green qPCR SuperMix, 0.4 $\mu$L of each primer (10 $\mu$M), 1 $\mu$L of cDNA, and 8.2 $\mu$L of nuclease-free water. The cycling conditions were 94°C for 30 s, followed by 45 cycles of denaturation at 94°C for 5 s and annealing at 60°C for 15 s, and the fluorescence signal was obtained at 72°C. The primers for target genes are listed in Table S2. The expression of target genes relative to the GAPDH gene was calculated using the $2^{-\Delta\Delta CT}$ method (51).

**Bacterial composition analysis.** Total DNA was extracted from fecal samples according to a bead-beating method (52). The integrity of bacterial DNA was detected via agarose gel electrophoresis, and the concentration and purity were detected with a Nanodrop 2000 instrument (Nanodrop, Wilmington, DE, USA) and Qubit3.0 spectrophotometer (Thermo Fisher Scientific, USA). The V3-V4 hypervariable regions of the 16S rRNA gene were amplified with the primers 341F (5′-CCTACGGGNGGCWGCAG-3′) and 805R (5′-GACTACHVGGGTATCTAATCC-3′) and then sequenced using an Illumina NovaSeq 6000 sequencer (Illumina, San Diego, CA, USA). The raw read sequences were processed in QIIME2 (53). The adaptor and primer sequences were trimmed using the cutadapt plugin. The DADA2 plugin was used for quality control and to identify amplicon sequence variants (ASVs) (54). Taxonomic assignments of representative ASV sequences were performed with a confidence threshold of 0.8 by a pretrained naive Bayes classifier which was trained on the RDP (version 11.5).

**Protein extraction and digestion, mass spectrometry, and data analysis.** The mucosal samples were homogenized with a MP FastPrep-24 homogenizer (MP Biomedicals, Illkirch-Graffenstaden, France) and then lysed in SDT lysis buffer (4% SDS, 100 mM dithiothreitol [DTT], 150 mM Tris-HCl pH 8.0). The lysates were further sonicated and boiled for 15 min. After centrifuged at 14,000 × $g$ for 40 min, the supernatant was quantified with the bicinchoninic acid (BCA) protein assay kit (Bio-Rad, Mississauga, ON, USA). Proteins from two pigs per group were pooled as a biological sample, and three biological replicates were obtained for each group. An appropriate amount of protein was taken from each sample and mixed into pooled samples for the construction of the spectral library. An SDS-PAGE assay was used to evaluate the extraction efficiency.

Protein digestion was performed using the filter-assisted sample preparation (FASP) method (55). All samples were digested in solution with trypsin. DTT was added to a final concentration of 20 mM, incubated at 37°C for 1 h, and then cooled to room temperature, and 100 $\mu$L iodoacetamide (100 mM) was added to a final concentration of 25 mM. The mixture was shaken at 600 rpm for 1 min and incubated at room temperature for 30 min, and 100 $\mu$L NH$_4$HCO$_3$ buffer (25 mM) was added to dilute the urea concentration to less than 1.5 M. Then, 40 $\mu$L NH$_4$HCO$_3$ buffer (25 mM) was added, and the mixture was shaken at 600 rpm for 1 min and incubated at 37°C for 2 h. Next, 4 $\mu$g trypsin was added to the sample, which was then incubated at 37°C for 16 h. Samples were reconstituted with 0.1% formic acid (FA) after being desalted and lyophilized. The peptide concentration was determined by UV light spectral density at 280 nm with a Nanodrop 2000c spectrophotometer (Thermo Fisher Scientific, Inc.). One hundred micrograms of low-abundance peptides after isolation was taken, and all fractions were collected.

The peptides were analyzed by liquid chromatography-tandem mass spectrometry (LC-MS/MS). Data-independent acquisition (DIA) analysis was performed using an EvoSep One system (EvoSep Inc., Odense, Denmark) for chromatographic separation. Peptides were separated with a binary solvent system consisting of solvent A (0.1% FA) and solvent B (0.1% formic acid acetonitrile). The samples were separated by nano-high-performance liquid chromatography (nano-HPLC) and analyzed by DIA mass spectrometry using a timsTOF mass spectrometer (Bruker Daltonics, Bremen, Germany). The detection mode was positive ion, and the MS and MS/MS range was 100 to 1,700 m/z. The MS2 uses the DIA data acquisition mode with 8 trapped-ion mobility spectrometry (TIMS) scan acquisition windows, each with an accumulation time of 100 ms. In the PASEF mode, the magnitude of the collision energy varies linearly with the ion mobility 1/$K$0 (collision energy from 20 to 59 eV corresponds to ion mobility of 1/$K$0, or 0.6 to 1.6 V·s/cm²).

The DIA data were processed using Spectronaut software (version 14.4.200727.47784) with the database used for library construction. The software parameters were set as follows: the retention time prediction type was set to dynamic iRT, interference on MS2 level correction was enabled, cross run normalization was enabled, and all results had to pass through the filtering parameter. The $Q$ value cutoff was set to 0.01 (equivalent to a false discovery rate [FDR] of <1%).

**Statistical analysis.** The percentage of cells that stained positive for TLR4, MyD88, the NF-$\kappa$B p65, IL-1$\beta$, IL-6, TNF-$\alpha$, TLR4, MyD88, NF-$\kappa$B, and D-lactate concentrations, the diamine oxidase activities in serum, and mucosal gene expression levels were analyzed by the independent-sample $t$ test using SPSS 25.0 statistical software (SPSS Inc., Chicago, IL, USA).

For microbial diversity, alpha diversity and beta diversity were analyzed with the R package vegan (https://github.com/vegandevs/vegan). Community composition was assessed using R 3.5.1. The heat

map of significantly altered genera was generated using Origin software 2019b (Origin, Northampton, MA, USA).

For bioinformatics analysis of the proteome, principal-component analysis (PCA) was performed by using SIMCA-P (version 14.1; Umetrics, Umea, Sweden), Venn diagrams, volcano plots, and box plots were generated using R software packages, and the pathway annotation was obtained from the KEGG database (https://www.genome.jp/kegg/).

**Data availability.** The raw reads were deposited into the NCBI Sequence Read Archive (SRA) database (accession number PRJNA796454).

## SUPPLEMENTAL MATERIAL

Supplemental material is available online only.

**TABLE S1**, DOCX file, 0.01 MB.

**TABLE S2**, DOCX file, 0.01 MB.

## ACKNOWLEDGMENTS

This project was financially supported by the Jiangsu Province Belt and Road International Cooperation Project (grant BZ2021009).

We thank the members of Key Laboratory for Veterinary Bio-Product Engineering and Institute of Veterinary Medicine of Jiangsu Academy of Agricultural Sciences for the *M. hyorhinis* challenge. We also would like to thank many students of Institute of Food Safety and Nutrition of Jiangsu Academy of Agricultural Sciences for the sample collection.

All authors designed and executed the study, and Yingying Zhang wrote the manuscript. All authors read and approved the final version of the article.

We declare that there is no conflict of interest.

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
