## [Reviewer comments · mSystems]

Dysbiosis of Gut Microbiota and Intestinal Barrier Dysfunction in Pigs with Pulmonary Inflammation Induced by *Mycoplasma hyorhinis* Infection

Yingying Zhang, Yuan Gan, Jia Wang, Zhixin Feng, Zhaoxin Zhong, Hongduo Bao, Qiyang Xiong, and Ran Wang

Corresponding Author(s): Yingying Zhang, Jiangsu Academy of Agricultural Sciences

Review Timeline:

Submission Date:	March 23, 2022
Editorial Decision:	May 2, 2022
Revision Received:	May 15, 2022
Accepted:	May 18, 2022

Editor: Marc Cook

Reviewer(s): Disclosure of reviewer identity is with reference to reviewer comments included in decision letter(s). The following individuals involved in review of your submission have agreed to reveal their identity: Lianqiang Che (Reviewer #2); Kristen Foust (Reviewer #3)

Transaction Report:

DOI: <https://doi.org/10.1128/mSystems.00282-22>

May 2, 2022

Dr. Yingying Zhang
Jiangsu Academy of Agricultural Sciences
Nanjing
China

Re: mSystems00282-22 (**Dysbiosis of Gut Microbiota and Intestinal Barrier Dysfunction in Pigs with Pulmonary Inflammation Induced by *Mycoplasma hyorhinis* Infection**)

Dear Dr. Yingying Zhang:

Thank you for submitting your manuscript to mSystems. We have completed our review and I am pleased to inform you that, in principle, we expect to accept it for publication in mSystems. However, acceptance will not be final until you have adequately addressed the reviewer comments.

Dr. Zhang and team. Thank you for thoughtfully addressing the previous comments. I apologize for the delay as it took some time to receive this round of comments.

Please see the latest reviewer's comments and address them, specifically in the methodology and if you can answer any of the additional concerns/questions in the discussion or limitations (if necessary).

Preparing Revision Guidelines

Sincerely,

Marc Cook

Editor, mSystems

Journals Department
Reviewer comments:

Reviewer #2 (Comments for the Author):

Thank you very much. Your comments have improved the manuscript effectively.

Reviewer #3 (Comments for the Author):

The present study addresses whether *M. hyorhinis* induced lung inflammation disrupts gut microbiota in pigs. The novelty of this study is very unique and well thought out. However, there are certain details that should be addressed as outlined below.

1. In the Results section Lines 89-106. IHC analysis of lung tissue sections showed an inflammatory response marked by increased TLR4/MyD88/NF κ b.
 - a. Should perhaps consider studying NF κ B signaling in lung fluid (BAL) and serum to further solidify results.
 - b. Since the pro-inflammatory cytokines (TNF- α , IL-6, and IL-1 β) were probed in serum, why not probe for these in the lungs? Gene expression of these cytokines are directly correlated to NF κ b signaling.
2. In the discussion section. There needs to be further explanation of how lung inflammation alters gut microbiota in pigs. There are obviously several things happening but what mechanism is driving these reactions?
3. In the Methodology section, Lines 295-299.
 - a. Did inoculation only take place on days 19, 20, and 21? According to what is written here, an intranasal dose was given on the last day of your treatments. This would elicit an acute inflammatory response in which stress from animal handling could also play a factor in high levels of inflammation. How long after dosing did slaughter occur? This should also be mentioned.
 - b. Intraperitoneal dosing took place on day 19 prior to intranasal(day21) and intravenous (day20). How could it be claimed that *M. hyorhinis* induced inflammation alters gut microbiota if dosing in the gut took place first?
 - c. I would suggest in further research to perform a more longitudinal study where dosing could take place over a few weeks give or take.
3. Perhaps there should be more control groups described as follows to show what differences are occurring
 - a. Control - saline only inoculation
 - b. IP - intraperitoneal inoculation only
 - c. IV - intravenous inoculation only
 - d. IN - intranasal inoculation only
 - e. IP, IV, and IN as you have done here

Comments to the author

The present study addresses whether *M. hyorhinis* induced lung inflammation disrupts gut microbiota in pigs. The novelty of this study is very unique and well thought out. However, there are certain details that should be addressed as outlined below.

1. In the Results section Lines 89-106. IHC analysis of lung tissue sections showed an inflammatory response marked by increased TLR4/MyD88/NFκB.

a. Should perhaps consider studying NFκB signaling in lung fluid (BAL) and serum to further solidify results.

b. Since the pro-inflammatory cytokines (TNF-α, IL-6, and IL-1β) were probed in serum, why not probe for these in the lungs? Gene expression of these cytokines are directly correlated to NFκB signaling.

2. In the discussion section. There needs to be further explanation of how lung inflammation alters gut microbiota in pigs. There are obviously several things happening but what mechanism is driving these reactions?

3. In the Methodology section, Lines 295-299.

a. Did inoculation only take place on days 19, 20, and 21? According to what is written here, an intranasal dose was given on the last day of your treatments. This would elicit an acute inflammatory response in which stress from animal handling could also play a factor in high levels of inflammation. How long after dosing did slaughter occur? This should also be mentioned.

b. Intraperitoneal dosing took place on day 19 prior to intranasal(day21) and intravenous (day20). How could it be claimed that *M. hyorhinis* induced inflammation alters gut microbiota if dosing in the gut took place first?

c. It would suggest in further research to perform a more longitudinal study where dosing could take place over a few weeks give or take.

3. Perhaps there should be more control groups described as follows to show what differences are occurring

- a. Control - saline only inoculation
- b. IP - intraperitoneal inoculation only
- c. IV - intravenous inoculation only
- d. IN - intranasal inoculation only
- e. IP, IV, and IN as you have done here

May 18, 2022

Dr. Yingying Zhang
Jiangsu Academy of Agricultural Sciences
Nanjing
China

Re: mSystems00282-22R1 (**Dysbiosis of Gut Microbiota and Intestinal Barrier Dysfunction in Pigs with Pulmonary Inflammation Induced by *Mycoplasma hyorhinis* Infection**)

Dear Dr. Yingying Zhang:

Thank you for thoroughly addressing the comments of all the reviewers that have improved clarity for the readers.

Your manuscript has been accepted, and I am forwarding it to the ASM Journals Department for publication. For your reference, ASM Journals' address is given below. Before it can be scheduled for publication, your manuscript will be checked by the mSystems production staff to make sure that all elements meet the technical requirements for publication. They will contact you if anything needs to be revised before copyediting and production can begin. Otherwise, you will be notified when your proofs are ready to be viewed.

Publication Fees:

We recognize that the video files can become quite large, and so to avoid quality loss ASM suggests sending the video file via <https://www.wetransfer.com/>. When you have a final version of the video and the still ready to share, please send it to mSystems staff at mSystems@asmusa.org.

For mSystems research articles, if you would like to submit an image for consideration as the Featured Image for an issue, please contact mSystems staff at mSystems@asmusa.org.

Sincerely,

Marc Cook
Editor, mSystems

Journals Department
Table S2: Accept

Table S1: Accept